# Feasibility of Arterial Spin Labeling Magnetic Resonance Imaging for Musculoskeletal Tumors with Optimized Post-Labeling Delay

**DOI:** 10.3390/diagnostics12102450

**Published:** 2022-10-10

**Authors:** Chien-Hung Lin, Tsyh-Jyi Hsieh, Yi-Chen Chou, Clement Kuen-Huang Chen

**Affiliations:** 1Department of Medical Imaging, Chi Mei Medical Center, Yongkang, Tainan 71004, Taiwan; 2Department of Radiology, Faculty of Medicine, College of Medicine, Kaohsiung Medical University, Kaohsiung 80708, Taiwan

**Keywords:** arterial spin labeling, magnetic resonance imaging (MRI), musculoskeletal tumor, tissue perfusion, tumor vascularity

## Abstract

Arterial spin labeling (ASL) magnetic resonance imaging (MRI) is used to perform perfusion imaging without administration of contrast media. However, the reliability of ASL for musculoskeletal tumors and the influence of post-labeling delay (PLD) have not been fully clarified. This study aimed to evaluate the performance of ASL with different PLDs in the imaging of musculoskeletal tumors. Forty-five patients were enrolled and were divided into a malignant group, a hypervascular benign group, a hypovascular benign group and a control group. The tissue blood flow (TBF) of the lesions and normal muscles was measured and the lesion-to-muscle TBF ratio and differences were calculated. The results showed that both the TBF of lesions and muscles increased as the PLD increased, and the TBF of muscles correlated significantly and positively with the TBF of lesions (all *p* < 0.05). The TBF and lesion-to-muscle TBF differences of the malignant lesions were significantly higher than those of the hypovascular benign lesions and the control group in all PLD groups (all *p* < 0.0125) and only those of the hypervascular benign lesions in the longest PLD (3025 ms) group (*p* = 0.0120, 0.0116). In conclusion, ASL detects high TBF in malignant tumors and hypervascular benign lesions, and a longer PLD is recommended for ASL to differentiate musculoskeletal tumors.

## 1. Introduction

Magnetic resonance imaging (MRI) is essential in the diagnosis of musculoskeletal tumors, providing detailed information before patients undergo surgery [1,2]. The T1- and T2-weighted images and chemical shift imaging differentiates tissue characteristics such as fat, fibrosis and chondroid components [1,2]. In addition, functional MRI, such as diffusion-weighted imaging, perfusion-weighted imaging, and magnetic resonance spectroscopy, provides more biomarker information [3,4,5,6]. Perfusion-weighted images are usually required to identify the vascularity and angiogenesis of tumors [7]. The most common perfusion-weighted imaging method is dynamic contrast-enhanced (DCE) imaging, which provides qualitative and quantitative assessments of in vivo hemodynamics [8,9,10].

However, DCE MRI necessitates the prior administration of a gadolinium-based contrast medium that may not be tolerated by some patients. To perform perfusion imaging without the use of a Gd-based contrast media, the arterial spin labeling (ASL) technique, a noninvasive MRI technique for measuring tissue perfusion, was recently developed [11,12]. The results of ASL sequencing are based on subtracting data from a control image and a “labeled” image using magnetically “labeled” protons in the blood flow by special radiofrequency pulse sequencing [11,12]. ASL is routinely used in the evaluation of cerebral blood flow, and some studies have applied the ASL sequence for other lesions, including head and neck tumors, breast tumors, renal tumors, portal venous circulation, prostate tumors, and multiple myeloma [13,14,15,16,17,18,19,20,21]. The findings of these studies suggest that the ASL results correlate highly with that of Gd-enhanced MRI. One recent study suggested that ASL is valuable for discriminating between benign, intermediate and malignant musculoskeletal tumors [22]. Most ASL studies, including a previous study of musculoskeletal tumors, have used single post-labeling delay (PLD), i.e., 1500–2000 ms, which is recommended for cerebral blood flow [23]. However, the blood flow velocity at the extremities is lower than that of the carotid artery [24,25]. The lower flow velocity may result in delayed enhancement in lesions at the extremities, and a PLD of 1500–2000 ms may not be optimal for perfusion study of the extremities. One previous study used a split-label design to improve measurements of muscle perfusion due to the long transit times in the extremities [26]. However, the influence of different PLDs in ASL perfusion imaging of musculoskeletal tumors has not been fully clarified Therefore, this study aimed to evaluate changes in the ASL-derived perfusion in different PLDs and to assess the performance of ASL in differentiating musculoskeletal tumors.

## 2. Materials and Methods

### 2.1. Subjects

A total of 45 consecutive subjects (19 female, 26 male) aged between 21 and 74 years who were treated at Chi Mei Medical Center between August 2017 and June 2020 were recruited. Inclusion criteria were (1) adults; (2) clinically suspicious mass lesion at lower extremities and (3) follow up examination after excision of previous tumor at lower extremities. Exclusion criteria were (1) having claustrophobia; (2) having contra-indication for MR examination; (3) pregnancy; and (4) incomplete MR examination. Subjects were divided into a malignant group, a hypervascular benign group, a hypovascular benign group and a control group based on the patients’ pathological reports or imaging findings. Subjects with a pathological diagnosis of malignant tumor were classified as the malignant group. The diagnoses of benign lesions were based on the pathological reports or the typical imaging findings. The imaging diagnosis criteria of these benign lesions were provided in the Appendix A [27,28,29,30,31]. The benign lesions with intense enhancement in the post-enhanced MRI were classified as the hypervascular benign group, such as schwannoma, hemangioma and fibrous dysplasia (Figure 1). The benign lesions without significant enhancement in the post-enhanced MRI were classified as the hypovascular benign group, such as lipoma and cysts. Subjects without significant mass lesions at visible muscles and bones were classified as the control group.

### 2.2. Ethical Considerations

The study was conducted according to the guidelines of the Declaration of Helsinki. The study protocol was approved by the Institutional Review Board of Chi Mei Medical Center (IRB Serial No.:10607-002, 12 July 2018). Signed informed consent was provided by all participants involved in the study.

### 2.3. MRI Acquisition

MRI acquisition was performed on a 3.0T MR scanner (MR750; GE Healthcare, Milwaukee, WI, USA) with an eight-channel phased-array torso coil, an ADW4.6 workstation and FuncTool^®^ software (GE Healthcare, Chicago, IL, USA). After tri-planar localizer scans were performed, routine MRI protocols including T1-weighted fast spin echo (FSE) sequence, T2-weighted FSE fat-suppressed sequence, diffusion-weighted imaging, dynamic contrast-enhanced T1-weighted FSE fat-suppressed sequence, and post-enhanced T1-weighted FSE fat-suppressed imaging were acquired. Subsequently, pseudo-continuous arterial spin labeling (pCASL) sequence with the aid of a 3D background suppressed fast spin-echo stack-of-spiral readout module was performed, and the imaging parameters were as follows: TR/TE = 4463/10.2 ms, labeling duration = 1500 ms, post-labeling delay (PLD) = 1025; 1525; 2025; 2525; 3025 ms, 512 acquisition points, six in-plane spiral interleaves, NEX = 3, and slice thickness = 4 mm.

### 2.4. Image Processing and Data Acquisition

All ASL data were exported to the manufacturer-supplied workstation (GE ADW 4.6). Post-processing to obtain the tissue blood flow (TBF) after slice-by-slice fusion with the T1-weighted images was performed at the same workstation by a radiologist (TJH) with 21 years of experience in musculoskeletal imaging. Regions of interest (ROIs) for quantitative analysis of lesions and normal muscles were manually placed on the TBF maps (Figure 2). In subjects with a hypervascular mass lesion noted in the post-enhanced T1W images, the ROIs were placed in the enhanced portion of the mass according to the post-enhanced images. In subjects with a mass lesion without significant enhancement, such as a cyst, the ROIs were placed in the mass lesion. In the mass-free subjects with lesions with abnormal enhancement, such as synovitis, the ROIs were placed in the regions with abnormal enhancement. In the subjects without mass or abnormal enhancement, the ROIs were placed in muscles. To obtain the TBF of normal muscles in the same slice, the muscles with normal morphology and signal intensities in T1-weighted, T2-weighted, and post-enhanced T1-weighted images were selected for placing the ROIs. The ROIs were measured three times with a same size in each target, and the mean value was calculated. The lesion-to-muscle TBF ratio (TBF ratio) was determined by dividing the lesion TBF by the muscle TBF. The lesion-to-muscle TBF difference (dif. TBF) was determined by subtracting the muscle TBF from the lesion TBF.

### 2.5. Statistical Analysis

All statistical analyses were performed using the JMP statistical software 16.0.0 (SAS Institute, Cary, NC, USA). Differences between the lesion TBF and the muscle TBF in the different PLD groups were evaluated using the paired *t*-test and *p* < 0.05 was considered statistically significant. Group differences in TBF were assessed using one-way analysis of variance followed by Tukey’s post hoc test to compare TBF between the four groups, and *p* < 0.0125 (0.05/4) was considered statistically significant. Group differences in PLD were assessed using one-way analysis of variance followed by Tukey’s post hoc test to compare TBF between the five groups, and *p* < 0.01 (0.05/5) was considered statistically significant. Pearson’s correlation coefficient analysis was performed to understand the relationship between the mean TBF values of lesions and muscles, and the results were considered statistically significant when the *p*-value was less than 0.05 (*p* < 0.05).

## 3. Results

The analytic sample data were from 45 subjects (mean age, 50.0 ± 14.0 years; range, 21–74 years), comprising 19 women (mean age, 53.8 ± 13.5 years; range, 23–74 years) and 26 men (mean age, 47.9 ± 14.8 years; range, 21–74 years). Nineteen subjects underwent surgery after the MRI examinations and the pathology reports showed seven malignant tumors (Table 1) and 12 benign lesions (Table 2). In the other subjects, mass-like lesions with typical benign MRI findings were noted in 17 subjects (Table 2), while no mass lesions were found in the MRIs of the other 9 subjects.

Based on the pathology reports or imaging findings, 7 subjects (2 women; mean age, 59.3 ± 12.7 years; range, 39–71 years) were included in the malignant group, 11 subjects (5 women; mean age, 46.9 ± 14.9 years; range, 23–71 years) in the hypervascular benign group, 18 subjects (9 women; mean age, 59.4 ± 13.9 years; range, 32–74 years) in the hypovascular benign group and 9 subjects (3 women; mean age, 46.8 ± 14.2 years; range, 21–74 years) in the control group. No significant differences were noted in age between the four groups (all *p* > 0.05).

All ASL images were performed successfully. However, high signal spiral artifacts around large arteries were noted in all ASL images with different PLD among the 10 subjects, including three in the hip region, five in the thigh region and two in the knee region (Figure 3). All conventional images of these subjects were free of motion artifacts and subjects denied any possibility of motion during their MRI examinations. Spiral artifacts were not noted in regions below the knee. The findings of the data of the patients without spiral artifacts were similar to those with all patients. Details of the data of the patients without spiral artifacts are provided in the Appendix A and the Appendix A.

Table 3 shows the mean TBF of lesions and normal muscles, the TBF difference and the TBF ratio in the different PLD groups. The TBF of lesions increased as the PLD increased in all 4 group (Figure 4a). The TBF of lesions in the malignant group was significantly higher than those of the hypovascular benign group and the control group in all PLD groups (all *p* < 0.0125). However, the TBF of lesions in the malignant group was significantly higher than that of the hypervascular benign group only in the highest PLD group (PLD = 3025 ms; *p* = 0.0120). No significant differences were found in the other PLD groups (all *p* > 0.0125).

The TBF of muscles also increased as the PLD increased in all four groups (Figure 4b). The mean TBF values of muscle in the hypervascular benign, hypovascular benign and control groups were similar. The mean TBF values of muscle of the malignant group were higher than those of the other three groups in all PLD groups but no significant differences were found between groups (all *p* > 0.0125).

The lesion-to-muscle TBF difference of the malignant group also showed the increasing trend as the PLD increased (Figure 4c). The lesion-to-muscle TBF difference of the malignant group was significantly higher than those of the hypovascular benign group and the control group in all PLD groups (all *p* < 0.0125). However, the lesion-to-muscle TBF difference of the malignant group was significantly higher than that of the hypervascular benign group only in the highest PLD group (PLD = 3025 ms; *p* = 0.0116), and no significant differences were found in other PLD groups (all *p* > 0.0125).

The lesion-to-muscle TBF ratio did not show a trend similar to that of the TBF of lesion and muscle (Figure 4d). No significant differences were shown in the lesion-to-muscle TBF ratio between the malignant group and the other 3 groups (all *p* > 0.0125).

Table 4 shows the differences between the lesion TBF and the muscle TBF in different groups and with different PLDs. In the malignant group, the lesion TBFs were significantly higher than those of the muscle TBFs in all PLD groups (all *p* < 0.05). In the hypervascular benign group, the lesion TBFs were significantly higher than those of the muscle TBFs in the groups with PLD = 1025 ms, 1525 ms, 2525 ms and 3025 ms (all *p* < 0.05). In the hypovascular benign group, the lesion TBFs were significantly lower than those in the muscle TBFs in the group with PLD = 2525 ms (*p* = 0.0336). In the control group, no significant differences were shown between the lesion TBFs and the muscle TBFs in all PLD groups (all *p* > 0.05).

In the 36 subjects with mass lesions, associations between the mean TBF of lesions and those of muscles in the different PLD groups are shown in Figure 5. Of note, the mean TBF of muscles was significantly and positively correlated with the mean TBF of lesions in all PLD groups (all *p* < 0.0001; r = 0.575–0.764).

## 4. Discussion

Previous ASL studies of musculoskeletal tumors used a single PLD that was recommended for brain perfusion. In the present study, multiple PLDs were compared relative to their ability to differentiate musculoskeletal tumors. The results of the present study showed that both TBF of lesions and muscles increased as the PLD and the TBF of muscles showed a significant positive correlation with the TBF of lesions. The TBF and lesion-to-muscle TBF differences of the malignant lesions were significantly higher than those of the hypovascular benign lesions and the control group in all PLD groups and those of the hypervascular benign lesion only in the longest PLD group (PLD = 3025 ms).

ASL is a non-invasive MRI perfusion method, and its merits include no contrast medium injection, repeated examination safely and quantitative values [32]. These merits make ASL suitable not only for routine brain perfusion examination but also for perfusion studies in other body parts [12,13,14,16,17,18,21,22,23,33]. Although many studies have been published in the musculoskeletal field, those studies have focused mainly on exercise and peripheral arterial disease [26,34,35,36]. Recently, a study investigating musculoskeletal tumors found that ASL was able to diagnose malignant tumors efficiently and effectively [22]. However, the benign tumor group in that study included hypervascular and hypovascular tumors and the mixed tumor types may actually result in lower TBFs, that may differ significantly from the malignant tumor group. In the present study, benign lesions were divided into hypervascular and hypovascular groups. The results showed that the TBFs of lesions in the malignant group were significantly higher than those of the hypovascular benign group and the control group in all PLD groups. Although the TBF of lesions in the malignant group was also higher than those of the hypervascular benign group, significant difference was noted only in the longest PLD group (PLD = 3025 ms). In Figure 4, the TBF of lesions in the hypervascular benign group showed nearly linear increase as PLD but those in the malignant group had a larger ascension slope in the longer PLD groups. The phenomenon of higher differences in the TBF of lesions between the malignant group and the hypervascular benign group in the longer PLD group may be due to longer transit times in the extremities than those in the brain. Compared with the PLD recommended for brain examination (PLD = 1500–2000 ms), the present study found that the differentiation of tumors was better with the longer PLD.

To obtain the related TBF data, the present study assessed the lesion-to-muscle TBF differences and ratio. The lesion-to-muscle TBF difference of the malignant group and the hypervascular benign group showed an increasing trend corresponding to increases in the PLD and that of the malignant group, which was significantly higher than those of the hypovascular benign group and the control group in all PLD groups. However, the lesion-to-muscle TBF difference of the malignant group was significantly higher than that of the hypervascular benign group only in the longest PLD group (PLD = 3025 ms) and the finding was similar to that of the TBF of lesions. The findings of both the TBF of lesions and lesion-to-muscle TBF difference suggests that a longer PLD, instead of the suggested PLD for brain perfusion (1500–2000 ms), should be considered in ASL evaluation of musculoskeletal tumors.

In contrast, the lesion-to-muscle TBF ratio did not show an increasing trend as the PLD increased, and no significant differences were found in the lesion-to-muscle TBF ratio between the malignant group and the other groups. The non-significant findings in the present study may be associated with the significant correlation between the TBF of lesions and normal muscles in subjects with mass lesions. These findings suggest that the lesion-to-muscle TBF ratio was not a useful value in differentiating between malignant and benign lesions.

Furthermore, the mean TBF values of muscle in the hypervascular benign, hypovascular benign and control groups were similar in the present study and were lower than those of the malignant group. In previous studies of musculoskeletal tumors, the perfusion MRI images, including dynamic contrast-enhanced MRI and ASL, were able to identify the hypervascularity of malignancy. However, the vascular changes of normal muscles adjacent to musculoskeletal malignancies has not been well explored. Tumor angiogenesis is important in the diagnosis and treatment of malignancy and the influence of adjacent normal tissue has also been reported [37]. In the present study, the morphology and signal intensities of normal muscles adjacent to the malignancy were acceptable but the malignancy group did not have increased TBF. Although the differences of TBF in normal muscles between the malignant group and the other groups were not statistically significant, comparison of the TBF of lesions and normal muscles showed significant correlation in all PLD groups. Results of the present study suggest that the perfusion change of normal muscle in the malignant group should be valued, especially in calculating the relative enhancement between tumor and muscle. Clearly, further study is needed.

In the present study, the high signal spiral artifacts in the ASL images were noted in 10 patients. The artifacts were noted in the hip, thigh, and knee regions and an intensely high flow of a large artery was noted in the center of the spiral artifacts in these ASL images. The similar artifacts mentioned in a previous brain ASL examination were shown to have resulted from severe motion [38]. However, all conventional images of those subjects were free of motion artifacts and the subjects denied any possibility of motion during MRI examination. Furthermore, the subjects with lesions in the regions below the knee did not have the spiral artifacts. The spiral artifacts were considered to be due to significant pulsation of the large arteries. To obtain a more accurate ASL measurement, further studies are needed involving improvement of these spiral artifacts.

The present study has a few limitations. Firstly, some patients with benign lesions were diagnosed only according to the images and the diagnoses may be not accurate completely. To minimize the impact to the results, the grouping had been made according to not only diagnosis but also enhancement patent. A further study with pathology-approved lesions might provide more information about individual tumors. Technically, ASL provides data of tissue perfusion without contrast administration. However, poor resolution and high background noise made small lesions undetectable in this study. Recent improvements in hardware and software, which produce a better signal-to-noise ratio, may improve estimation. The small sample size is a limitation affecting the power of this study and only nine patients with malignant tumors participated in the ASL measurements. Furthermore, the types of tumors are heterogenous and many of these are represented by a single case. An extended prospective study with a large number of target tumors may provide more information about the application of ASL for differentiating musculoskeletal tumors.

## 5. Conclusions

ASL MRI detects high TBF of malignant tumors and hypervascular benign tumors. The TBF and lesion-to-muscle TBF differences can be used to differentiate malignant tumors and hypervascular benign lesions by using the longest PLD (PLD = 3025 ms) but not the common PLD used for brain perfusion. ASL with a longer PLD is recommended for better detection of malignant tumors.

## Figures and Tables

**Figure 1 diagnostics-12-02450-f001:**
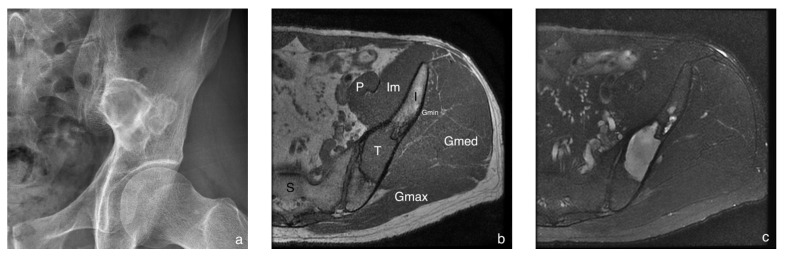
A 37-year-old man with a fibrous dysplasia at left ilium. The radiography (**a**) shows an osteolytic lesion with sclerotic margin and some intra-lesion sclerosis. The axial sections of MRI (**b**–**d**) visualize a well-defined mass lesion at left ilium and the lesion shows homogeneously intermediate signal intensity on a T1-weighted sequence (**b**), homogeneous hyperintensity on a T2-weighted sequence without fat suppression (**c**), and homogeneously intense enhancement on a post-contrast T1-weighted sequence with fat suppression (**d**). The TBF maps (**e**) showed high TBF at the lesion (white arrows). (T: tumor; I: ilium; P: psoas muscle; Im: iliacus muscle; Gmin: gluteus minimus muscle; Gmed: gluteal medius muscle; Gmax: gluteus maximus muscle).

**Figure 2 diagnostics-12-02450-f002:**
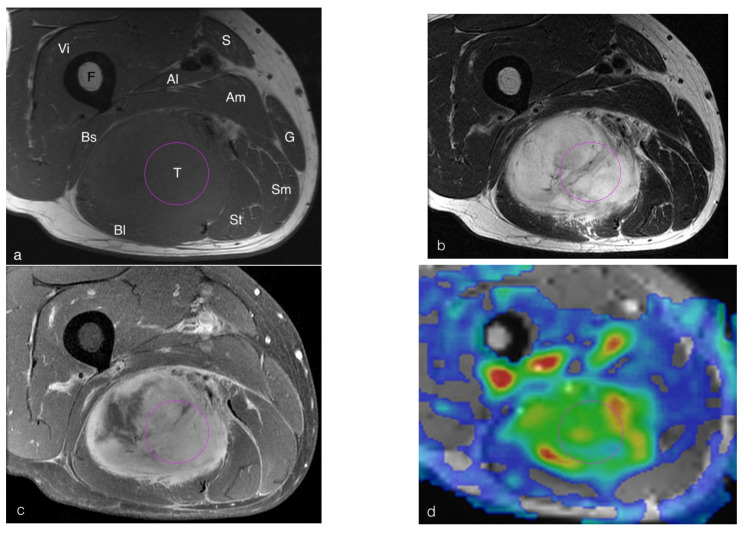
A 57-year-old man with a myxofibrosarcoma at right thigh. The axial section of a T1-weighted sequence (**a**), a T2-weighted sequence (**b**), and a post-contrast T1-weighted sequence with fat suppression (**c**) visualizes a mass lesion at the posterior compartment of the right thigh. The regions of interest were manually placed in the lesion and normal muscle on the TBF maps (**d**) according to the post-contrast T1-weighted image with fat suppression (**c**). (T: tumor; F: femur; Vi: vastus intermedius muscle; S: sartorius muscle; Al: adductor longus muscle; Am: adductor magnus muscle; G: gracilis muscle; Sm: semimenbranesus muscle; St: semitendinosus muscle; Bl: biceps femoris long head; Bs: biceps femoris short head).

**Figure 3 diagnostics-12-02450-f003:**
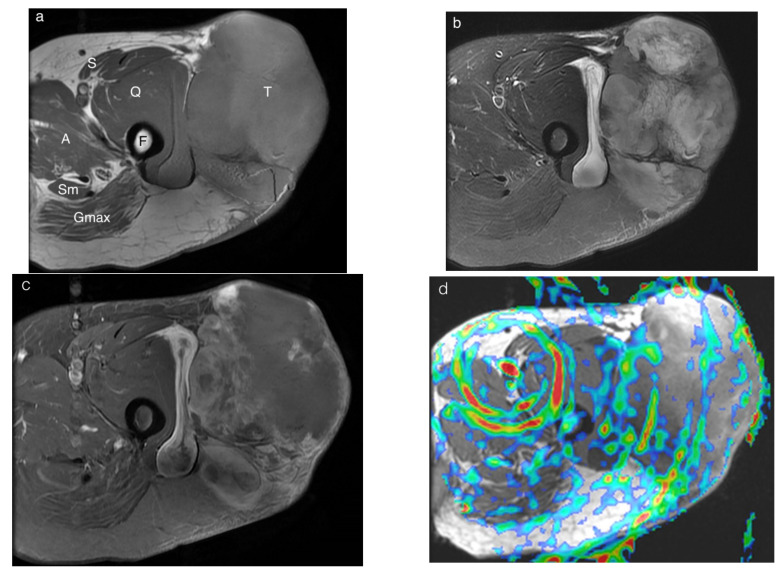
A 71-year-old woman with a myxofibrosarcoma at left upper thigh. The axial section of a T1-weighted sequence (**a**), a T2-weighted sequence without fat suppression (**b**), and a post-contrast T1-weighted sequence with fat suppression (**c**) visualizes a mass lesion at lateral aspect of the left upper thigh. The TBF maps (**d**) showed high signal spiral artifacts. (T: tumor; F: femur; S: sartorius muscle; A: adductor muscles; Sm: semimenbranesus muscle; Q: quadriceps femoris muscle; Gmax: gluteus maximus muscle).

**Figure 4 diagnostics-12-02450-f004:**
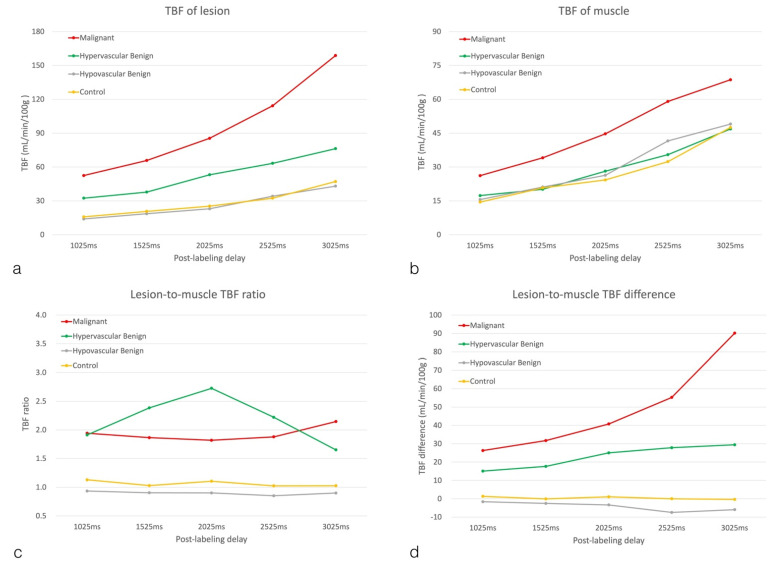
The mean TBF values of lesions (**a**) and normal muscles (**b**), the lesion-to-muscle differences (**c**) and the lesion-to-muscle TBF ratios (**d**) of the malignant group, the hypervascular benign group, the hypovascular benign group and the control group in different post-labeling delay times.

**Figure 5 diagnostics-12-02450-f005:**
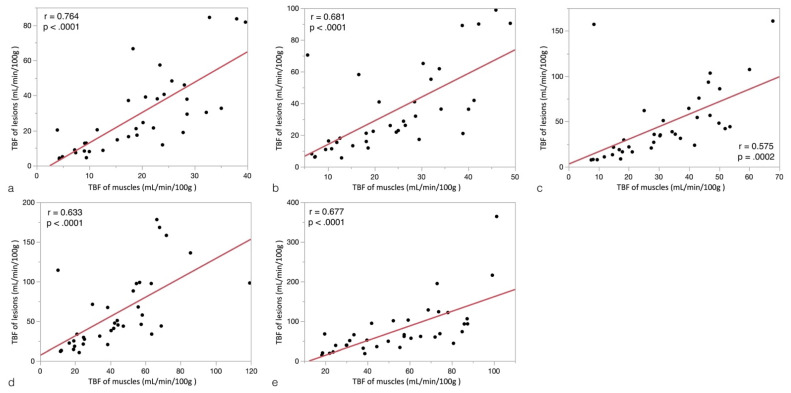
Associations between the mean TBF values of lesions and muscles in the different PLD groups (**a**): PLD = 1025 ms; (**b**): PLD = 1525 ms; (**c**): PLD = 2025 ms; (**d**): PLD = 2525 ms; (**e**): PLD = 3025 ms).

**Table 1 diagnostics-12-02450-t001:** Details of pathological diagnosis of the malignant group.

Diagnosis	N	Location (N)
Atypical lipomatous tumor (borderline malignancy)	1	Left thigh (1)
Liposarcoma, well-differentiated	1	Left thigh (1)
Pleomorphic rhabdomyosarcoma	1	Left thigh (1)
Myxofibrosarcoma	2	Left thigh (1) Right thigh (1)
Lymphoma	1	Right iliac bone (1)
Metastasis	1	Right iliac bone (1)

**Table 2 diagnostics-12-02450-t002:** Details of pathological and imaging diagnosis of the benign groups.

Diagnosis	N	Method	Location (N)
Hypervascular benign lesion	11		
Schwannoma	6	Pathology	Right hip (2) Left thigh (1) Right knee (1)
		MRI	Right leg (2)
Hemangioma	3	Pathology	Right foot (1)
		MRI	Right lower leg (2)
Fibrous dysplasia	2	MRI	Left femur (1) Right femur (1)
Hypovascular benign lesion	18		
Intramuscular myxoma	1	Pathology	Right hip (1)
Lipoma	6	Pathology	Right gluteal region (1)
		MRI	Right thigh (2) Left thigh (3)
Cyst	5	MRI	Left hip (1) Left foot (1) Left knee (2) Right ankle (1)
Enchondroma	1	MRI	Right tibia (1)
Necrosis	1	Pathology	Right thigh (1)
Synovial chondromatosis	2	Pathology	Right knee (1) Left knee (1)
Amyloidosis	1	Pathology	Left lower leg (1)
Gouty arthritis	1	Pathology	Left knee (1)

**Table 3 diagnostics-12-02450-t003:** TBF of lesions and normal muscles and lesion-to-muscle TBF difference and ratio in the post-Labeling delay of 1025ms to 3025 ms and the *p* value between the malignant, hypervascular benign, hypovascular benign and control groups.

	Malignant	Hypervascular Benign	Hypovascular Benign	Control	M-B	M-L	M-N	B-L	B-N	L-N
PLD = 1025ms										
Lesion TBF (mL/min/100 g)	52.46 ± 25.41	32.42 ± 25.03	13.96 ± 9.41	15.81 ± 9.39	0.1166	0.0004 *	0.0008 *	0.0524	0.0947	0.9933
Muscle TBF (mL/min/100 g)	26.18 ± 8.12	17.34 ± 10.50	15.57 ± 10.28	14.49 ± 7.66	0.2422	0.1181	0.0716	0.9631	0.8653	0.9909
Dif_TBF (mL/min/100 g)	26.28 ± 18.29	15.08 ± 17.19	−1.62 ± 3.73	1.32 ± 6.11	0.2429	0.0001 *	0.0007 *	0.0050 *	0.0265	0.9224
TBF ratio	1.94 ± 0.47	1.91 ± 1.26	0.93 ± 0.24	1.13 ± 0.39	0.9998	0.0432 *	0.1383	0.0092 *	0.0503	0.9079
PLD = 1525 ms										
Lesion TBF (mL/min/100 g)	65.76 ± 29.18	37.79 ± 28.51	18.60 ± 11.35	20.69 ± 10.20	0.0374	0.0002 *	0.0003 *	0.0898	0.1536	0.9935
Muscle TBF (mL/min/100 g)	34.08 ± 11.73	20.14 ± 11.35	21.12 ± 12.64	20.76 ± 9.67	0.0765	0.1113	0.0973	0.9961	0.999	0.9998
Dif_TBF (mL/min/100 g)	31.69 ± 18.43	17.65 ± 23.36	−2.52 ± 5.32	−0.07 ± 7.51	0.2473	0.0002 *	0.0006 *	0.0075 *	0.0224	0.9756
TBF ratio	1.86 ± 0.40	2.38 ± 3.08	0.90 ± 0.22	1.03 ± 0.29	0.9233	0.6574	0.7467	0.1291	0.1858	0.9976
PLD = 2025 ms										
Lesion TBF (mL/min/100 g)	85.46 ± 44.15	53.16 ± 43.84	22.95 ± 13.56	25.34 ± 13.32	0.1474	0.0008 *	0.0012 *	0.0657	0.1022	0.9970
Muscle TBF (mL/min/100 g)	44.69 ± 14.24	28.14 ± 15.94	26.33 ± 15.86	24.27 ± 13.20	0.1290	0.0769	0.0405	0.9896	0.9114	0.9850
Dif_TBF (mL/min/100 g)	40.77 ± 30.41	25.01 ± 41.51	−3.37 ± 6.63	1.07 ± 7.19	0.5951	0.0057 *	0.0118 *	0.0332	0.0925	0.9699
TBF ratio	1.82 ± 0.10	2.73 ± 4.83	0.90 ± 0.24	1.10 ± 0.28	0.8963	0.8926	0.9455	0.2999	0.4032	0.9972
PLD = 2525 ms										
Lesion TBF (mL/min/100 g)	114.25 ± 54.42	63.27 ± 45.89	34.12 ± 23.23	32.39 ± 16.58	0.0251	0.0002 *	0.0001 *	0.1603	0.1251	0.9993
Muscle TBF (mL/min/100 g)	59.04 ± 17.76	35.47 ± 20.70	41.57 ± 29.15	32.38 ± 15.99	0.1509	0.3881	0.0841	0.8949	0.9842	0.7147
Dif_TBF (mL/min/100 g)	55.21 ± 42.06	27.80 ± 35.66	−7.45 ± 11.20	0.02 ± 6.95	0.1409	<0.0001 *	0.0004 *	0.0051 *	0.0371	0.8748
TBF ratio	1.88 ± 0.59	2.22 ± 2.74	0.85 ± 0.20	1.02 ± 0.22	0.9673	0.5171	0.6610	0.1104	0.1953	0.9912
PLD = 3025 ms										
Lesion TBF (mL/min/100 g)	158.80 ± 109.78	76.27 ± 52.69	43.10 ± 23.41	47.13 ± 26.76	0.0120 *	0.0003 *	0.0004 *	0.3669	0.4807	0.9971
Muscle TBF (mL/min/100 g)	68.67 ± 17.70	46.86 ± 24.51	49.03 ± 24.49	47.50 ± 25.60	0.2731	0.3625	0.2979	0.9957	0.9999	0.9985
Dif_TBF (mL/min/100 g)	90.14 ± 92.73	29.41 ± 34.34	−5.93 ± 12.34	−0.37 ± 11.61	0.0116 *	<0.0001 *	0.0001 *	0.1047	0.2139	0.9826
TBF ratio	2.15 ± 0.87	1.65 ± 0.70	0.90 ± 0.26	1.03 ± 0.21	0.2362	0.0001 *	0.0005 *	0.0036 *	0.0196	0.9247

M-B = comparison between the malignant group and hypervascular benign group. M-L = comparison between the malignant group and hypovascular benign group. M-N = comparison between the malignant group and control group. B-L = comparison between the hypervascular benign group and hypovascular benign group. B-N = comparison between the hypervascular benign group and control group. L-N = comparison between the hypovascular benign group and control group. * *p* < 0.0125 indicates a significant difference between the malignant, hypervascular benign, hypovascular benign and control groups.

**Table 4 diagnostics-12-02450-t004:** Paired *t*-test for difference of the mean TBF values of lesions and muscles in four groups with different PLD.

	Malignant	Hypervascular Benign	Hypovascular Benign	Control
PLD	*p*	*p*	*p*	*p*
1025 ms	0.0169 *	0.0082 *	0.1439	0.4509
1525 ms	0.0084 *	0.0184 *	0.1131	0.9721
2025 ms	0.0219 *	0.0506	0.0916	0.5998
2525 ms	0.0236 *	0.0157 *	0.0336 *	0.9929
3025 ms	0.0315 *	0.0094 *	0.1088	0.9105

* *p* < 0.05 indicates a significant difference between the mean TBF values of lesions and muscles.

## Data Availability

The datasets generated during and/or analyzed during the current study are available from the corresponding author on reasonable request.

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
