# Peer review of "Feasibility of Arterial Spin Labeling Magnetic Resonance Imaging for Musculoskeletal Tumors with Optimized Post-Labeling Delay"

_diagnostics, 2022, doi:10.3390/diagnostics12102450_

Round 1

Reviewer 1 Report

The following is the summary of the present manuscript:

This study aimed to evaluate the performance of ASL with different PLDs in the imaging of musculoskeletal tumors. Forty-five patients were enrolled and were divided into a malignant group, hypervascular benign group, hypovascular benign group and a control group. The tissue blood flow (TBF) of the lesions and normal muscles were measured and the lesion-to-muscle TBF ratio and differences were calculated. Results showed that both the TBF of lesions and muscles increased as the PLD increased, and the TBF of muscles correlated significantly and positively with the TBF of lesions. In conclusion, ASL detects high TBF in malignant tumors and hypervascular benign lesions, and a longer PLD is recommended for ASL in differentiating musculoskeletal tumors.

   I think the article is well written and deserves be published. I have some minor suggestions:

1. In line 68, why did the authors only choose the tumors over the lower extremities?

Please clarify the reason.

2. In Figure 1, it would be helpful if the authors can add some annotations for the anatomic structure. The same comment can be applied for other MRI figure.

3. Did the authors use any statistical method to confirm that their data were normally distributed before applying the parametric test for comparison?

4. In Table 2, some tumors were defined by MRI. It would be helpful if the authors can provide a supplementary table to elaborate the criteria to diagnose those tumors.

Reviewer 2 Report

I have a major concern in regards to the current interesting research: Not all the patients have a confirmation of diagnosis with pathology. Underline this concern in limitations of the study ('discussion' section).

Moreover, since many patients have only an imaging (MRI) 'supposed' diagnosis you have to state which imaging features have been used, which diagnosis can be made (supposed with a certain precision) directly with MR? (Cysts, hemangioma/MAV, ossificans myositis, schwannoma/neurinoma...). Discuss this in discussion, and, most importantly prepare a paragraph of methods detailing this issue. This is very important, and need to be supported by some literature references. I suggest You these general references (for soft tissue PMID: 25344599 - PMID: 23673547 - PMID: 25256016) - (for bone PMID: 35236589 - PMID: 12700923); Moreover, for hemangioma in regards to the possibility of imaging diagnosis avoiding biopsy, you can use this reference (PMID: 29476440).

Moreover, I suggest You improve the abstract including the main results (%) with main statistical data (p-values).

The iconographic part is focused on soft tissue only, please provide 2 figures including a couple of bone tumors as examples.

Did you exclude patients with spiral artifacts from the research? Discuss and detail this point

Round 2

Reviewer 2 Report

I am satisfied with the revisions performed, I have only a minor concern

In the Figure 1 (added in revision) you state that high TBF inside the lesion is registered (panel e). Many artifacts seem to be present, at least inside the pelvis, please use arrows or some other modality to better depict this TBF increase in the figure.

Author Response

Response: Thank you for the comments. The white arrows have been added in the Figure 1e to show the location of the tumor and the changes of the legend of Figure 1 have been made as shown below:

Revised figure and legend:

The TBF maps (e) showed high TBF at the lesion (white arrows).
